

# An AutoEncoder enhanced light gradient boosting machine method for credit card fraud detection

Lianhong Ding[1], Luqi Liu[1], Yangchuan Wang[1], Peng Shi[2] and Jianye Yu[1]

[1] Beijing Wuzi University, Beijing, China
[2] University of Science and Technology Beijing, Beijing, China

## ABSTRACT

Online financial transactions bring convenience to people's lives, but also present vulnerabilities for criminals to embezzle users' accounts and trick users into credit card fraud. Although machine learning methods have been adopted to detect anomalous transactions, it's hard for a single machine learning method to achieve satisfying results with the increasing scale and dimensionality of financial datasets. In addition, for anomaly detection of financial data, there is an obvious imbalance between normal records and abnormal. In this situation, the experimental results cannot be objectively evaluated only by the traditional metrics, such as precision, recall, and accuracy. This paper proposes an AutoEncoder enhanced LightGBM method for credit card detection. The method inherits the advantages of each component, using an AutoEncoder for feature reconstruction on the dataset, and integrating the LightGBM algorithm for improving the GBDT (Gradient Boosting Decison Tree) to detect abnormal data more accurately and efficiently. Besides the traditional evaluation metrics, F-measure, area under curve (AUC), Matthew's correlation coefficient (MCC), and balanced classification rate (BCR) are also adopted as the evaluation metrics. Two financial datasets were used to validate the performance and robustness of the proposed model. Results obtained from the credit card fraud dataset containing 31 features indicate that our model significantly outperforms other models with a recall of 94.85%, representing a 10.70% improvement compared to the best detection performance model with a recall of only 86%. Additionally, our model's BCR score is also significantly better than other models, with a BCR score of 97%, as opposed to the best detection performance model's BCR score of 92%, representing a 5% improvement by our model. Various sampling methods and model combinations were considered in this study. It was found that the SMOTE algorithm combined with the proposed model produced the best results, with an AUC value of 96.83% and an F-measure score of 80.27%. The Santander bank transaction record dataset is a high dimensional large dataset containing 200 features. Experimental results on this dataset reveal that compared to other models, our model significantly improved recall and F-measure results, raising the recall to 94.14% and the F-measure score by 11.51%, surpassing the second-best-performing model. Overall, these findings demonstrate the robustness and superiority of our model in detecting fraudulent transactions and highlight the effectiveness of the SMOTE algorithm in combination with the proposed model.

Corresponding author
Jianye Yu, yujianye@bwu.edu.cn

# INTRODUCTION

The continuous innovation and breakthroughs of Internet banking have made online payment more and more convenient. Compared to traditional bank card payment, online payment offers simpler transaction procedure due to its convenience and immediacy. At the same time, the demand for efficient data processing and analyzing is growing accordingly. Unfortunately, the convenience brought by online payment also exposes the society and economy system to the risk of financial crime. Financial fraud not only harms people's vital interests, but also brings about huge economic losses and major negative effects to financial institutions. Therefore, timely and effective identification of anomalies in financial data can not only ensure the safety of users' assets, but also benefit the sustainable development of the financial industry.

Since the characteristics of online virtual payment account transactions, whether they are related to credit card or not, are similar with credit card transaction, this paper mainly focuses on the credit card fraud detection. From the point of view of data analysis, credit card fraud detection is essentially an anomaly detection problem. In the field of data mining, the purpose of anomaly detection is to identify items or events that do not match the expected pattern or other items in the dataset. The concept of "anomaly" was first given by *Grubbs (1969)*: "Outliers, that is, points that deviate significantly from other members of the sample in comparison." According to the form of anomalies actually detected, anomalies can be divided into point anomaly, collective anomaly, and context anomaly (*Chandola, Banerjee & Kumar, 2009*). As for the credit card fraud detection problem studied in this paper, we need to identify abnormal transaction records from the actual credit card transaction dataset to minimize the economic losses caused by credit card transaction fraud to cardholders and enterprises.

At present, many researchers have conducted research on this issue and proposed many corresponding methods. As early as the 19th century, the statistical community has studied the detection of outliers or anomalies (*Edgeworth, 1887*). The general idea of the statistical method of anomaly detection is that the normal data is assumed to be produced by a statistical model, and the data that does not conform to that model is abnormal. But the effectiveness of statistical methods is highly dependent on the validity of the statistical model assumptions made about the given data. With the development and application of database technology, many researchers have appied data mining methods to the problem of intrusion detection (*Caulkins, Lee & Wang, 2005*). However, the normal data mining analysis methods cannot handle a large number of multiple types of data features or variables well, and the amount of calculation is increasing significantly. The rise of machine learning has enabled machine learning technology to be widely used in credit card fraud detection problems. Existing machine learning based credit card fraud detection methods are mainly divided into two categories: unsupervised methods and supervised

methods (*Ahmed, Mahmood & Islam, 2016*). But they suffer from a low detection accuracy and not able to solve the high dimensional. This is mainly due to the following problems:

1. There is an imbalance on the dataset, which means the normal data is the majority while the abnormal data is the minority. The prediction result is biased towards the category which has larger proportion.

2. Feature selection plays a crucial role in the performance of models for credit card fraud detection. Currently, feature selection for credit card fraud detection is typically performed manually using methods such as rule-based, knowledge-based, and statistical modeling approaches. However, these methods often fail to effectively uncover underlying patterns in data, resulting in difficulties in feature selection and impacting model performance.

In this paper, we use the synthetic minority oversampling technique (SMOTE) method to balance the raw data, and use unsupervised anomaly detection method AutoEncoder (AE) to enhance the features. After the features are enhanced, the LightGBM was used for detection (AEELG). In addressing the challenge of imbalanced datasets, the selection of an appropriate sampling method should take into consideration the data distribution and its integration performance with the model. Through experiments, we have discussed the effects of various sampling methods when they are combined with our model. Among them, SMOTE has shown the best integration performance with our model. In addressing the challenge of feature selection, we used AutoEncoder for feature reconstruction offers notable advantages compared to feature selection approaches based on statistics or machine learning. The AutoEncoder can automatically learn the optimal feature representation from the input data without relying on labeled training data. This characteristic eliminates the dependence on annotated data and reduces the burden of data labeling, especially when dealing with large-scale datasets. Furthermore, the AutoEncoder employs a layer-by-layer encoding and decoding process to acquire deep features. This enables the model to uncover latent structures and patterns within the data, ultimately capturing more comprehensive and informative features. Two imbalanced datasets publicly available on Kaggle were employed. The first dataset contains 284,807 anonymous transactions. Only 0.172% of the dataset are positive samples. The second dataset comprises 200,000 anonymous transactions with 10.049% positive samples. Compared with other single and hybrid methods in references, our method performs better in various metrics.

The main contributions of this paper are summarized as follows:

1. AEELG is proposed where, the deep neural network AutoEncoder is adopted to reconstruct features to enhance the integrated algorithm LightGBM which can detect abnormal data from reconstructed datasets.

2. Seven different metrics are adopted to evaluate the experimental results on imbalanced datasets, which can evaluate experimental results more comprehensively.

3. AEELG exhibits a good classification performance and prediction capability on datasets with feature dimensions of both 31 and 200, indicating its excellent robustness. On the dataset with 31 features, AEELG achieves a recall of 94%, outperforming other twenty-two compared algorithms, which demonstrates that AEELG is superior sensitivity in detecting positive samples. AEELG also exhibits a better anomaly detection ability,

getting a BCR value of 97%. It is significantly higher than other algorithms. The balanced handling of positive and negative samples suggests that AEELG can accurately detect true anomaly data points while reducing false positives to the lowest level, providing more reliable anomaly detection results. On the dataset with 200 features, AEELG shows significantly higher AUC, recall, F-measure score, and MCC values than other algorithms. Such performance indicates that AEELG possesses exceptional accuracy, stability, and reliability in anomaly detection.

4. In order to find appropriate approach for imbalanced dataset handling in credit card fraud detection, nine sampling methods are compared when they are combined with AEELG. They are two over sampling methods, five undersampling methods, and one mixed sampling method. Experimental results reveal that both SMOTE and Borderline-SMOTE, an improvement of SMOTE, are good choices for sampling when they are combined with AEELG.

The rest of this paper is organized as follows. 'Related work' summaries the related works, pointing out their shortcomings on financial fraud detection. 'Methods' introduces the proposed AutoEncoder enhanced LightGBM (AEELG) method. 'Experiment' details the dataset, evaluation metrics and the experiment results. 'Conclusion' draws the conclusion and discusses the future work.

## RELATED WORK

Credit cards originated in the United States in 1915 and quickly became popular in the United Kingdom, Japan, Canada and European countries in the 1960s. With the rapid growth of customers' credit card holdings, the problem of credit card fraud has become increasingly obvious. Especially since the outbreak of the subprime financial crisis in 2008, the fatality rate and bad debt rate of credit cards have become higher and higher, making the identification and prevention of credit card fraud urgent.

Since there are some potential correlations among credit card consumption records, many researchers tried to identify fraud by data analysis methods. A method that utilizes decision trees and Boolean logic functions, combined with cluster analysis and consumption pattern detection, is used to distinguish the legitimacy of customers' credit card transactions (*Kokkinaki, 1997*). Apriori is modified and applied into a fraud pattern mining algorithm to achieve the purpose of mining credit card fraud data (*Chiu & Tsai, 2004*). An improved agglomerative hierarchical clustering method, achieved satisfactory results on the tensile test, HTRU2, and credit card datasets (*Shi et al., 2021*). Moreover, some researchers have also obtained many results in the field of credit card fraud detection using neural network methods (*Ghosh & Reilly, 1994*; *Brause, Langsdorf & Hepp, 1999*; *Akhilomen, 2013*; *Esenogho et al., 2022*; *Roseline et al., 2022*; *Zhou, Xue & Xu, 2022*). In this regard, *Zhang et al. (2021)* proposed a feature engineering framework based on homogeneity oriented behavior analysis (HOBA) to generate feature variables for the fraud detection. Then, deep learning techniques are incorporated into the fraud detection system to deliver good fraud detection performance. Similarly, a novel network-based credit card fraud detection method, CATCHM, enhanced efficiency through an innovative network

design, which includes an efficient inductive pooling operator and a carefully configured downstream classifier (*Van Belle, Baesens & De Weerdt, 2023*). However, with the increase in financial data and the progressively intricate data structure, the neural network also grows more intricate. Complex neural network structures can improve model performance and process more complex data, but they also face challenges such as increased training costs, increased risk of overfitting, and difficulty in parameter adjustment.

As credit card transaction data becomes more dynamic, massive, and high-dimensional, the calculation accuracy and speed of a single model are no longer sufficient to solve the problem of credit card fraud detection. In recent years, the integrated method represented by the boosting method in credit card fraud identification has been recognized by more and more scholars in academic scientific research and industry. Among them, Adaboost method is replaced by the AdaCost method because AdaCost can obtain the same recognition effect at a lower computational cost (*Chan et al., 1999*). Also, the GBDT algorithm, an additive boosting algorithm, is proposed (*Friedman, 2001*). It aims at the previous fit the next tree in the direction in which the residual of the tree decreases the fastest, so that the residual is quickly minimized, and finally all the obtained gradient boosting trees can be summed. Furthermore, the XGBoost algorithm was developed and applied in the engineering field based on the GBDT algorithm (*Chen & Guestrin, 2016*). On this basis, XGBoost algorithm is transformed into LightGBM algorithm after multiple optimizations (*Ke et al., 2017*). Experiments on multiple datasets show that the training speed of the LightGBM algorithm is faster than that of XGBoost with almost the same effect. In recent years, the improvement of the LightGBM algorithm has become a hot topic. Bayesian optimization is adopted to search best parameters for LightGBM, such as learning rate, max tree depth, and leaf number. It brings a higher accuracy in credit card detection experiments (*Huang, 2020*). LightGBM is also widely used for feature selection (*Sarıkaya, Günel Kılıç & Demirci, 2022*). Firstly, one LightGBM model is used for feature selection. Then, features are evaluated by another LightGBM classifier to determine the optimal feature set. Finally, a gated recurrent unit is adopted for attack detection by a recurrent neural network model. With the similar idea, the LightGBM algorithm is used for feature selection and an AutoEncoder is adopted for training to detect network intrusion from the NSL-KDD dataset (*Tang, Luktarhan & Zhao, 2020*). In addition, some scholars arranged and combined the features, and used genetic algorithms (GA) to select the best features (*Saheed et al., 2020*). However, these feature selection methods all rely on labeled data. Also, the feature selection method based on LightGBM usually only considers the importance of a single feature, while ignoring the relationship between features and combination effects. In addition, the use of GA to combine features is affected by the convergence of the algorithm and the limitation of search space.

Another major challenge in credit card transaction fraud detection is the imbalance of datasets. It is caused by the fact that criminals need to deviate from conventional system when carrying out criminal activities. Various methods have been proposed to balance imbalanced datasets in credit card transaction fraud detection, including sampling techniques, cost-sensitive learning, ensemble-based methods, and deep learning models. The core idea of sampling techniques is to adjust the number of samples for different

categories by increasing or reducing certain class samples. Balancing datasets using sampling techniques can greatly improve the performance of machine learning algorithms in predicting minority classes. Two popular over-sampling methods are synthetic minority over-sampling technique (SMOTE) and adaptive synthetic sampling (ADASYN). SMOTE works by creating synthetic samples of the minority class along the line segments that connect existing minority class samples, and *Mqadi, Naicker & Adeliyi (2021)* has shown its effectiveness in balancing financial datasets. ADASYN generates more synthetic samples in regions where the density of the minority class is lower, and *Abd El-Naby, Hemdan & El-Sayed (2022)* has demonstrated its positive results in credit card fraud detection. Cost-sensitive learning assigns different misclassification costs to different classes to improve model performance on the minority class. Techniques such as cost-sensitive decision trees (*Moral-García et al., 2022*) and cost-sensitive support vector machines (*Rezvani & Wang, 2022*) also are used to deal with the imbalance of datasets. Ensemble-based methods combine multiple weak classifiers to form a strong classifier and they have been proven to be effective for imbalanced classification problems. Methods such as AdaBoost (*Petrovic et al., 2022*) and Random Forests (*Dileep, Navaneeth & Abhishek, 2021*) have been widely used for credit card fraud detection. Recently, deep neural networks have been adopted to handle the imbalanced data problem in credit card fraud detection (*Al-Shabi, 2019*; *Alharbi et al., 2022*). These models can learn more discriminative representations of the minority class by reducing the impact of noisy and irrelevant features.

Although the above methods have achieved a certain effect in different ways, there are still shortcomings. The structure of the single model like neural network will become more and more complex with the scale of the dataset larger and larger. Moreover, in terms of feature selection, existing feature selection methods based on statistical approaches and machine learning solely consider the individual importance of features, thereby disregarding the interrelationships among features, which can influence the ultimate classification result. The use of GA to combine features subject to the limitations of algorithm convergence and the search space, possibly leading to the inability to discover globally optimal feature combinations. Additionally, in the field of boosting method, their evaluation system for experimental results is not comprehensive enough to further improve various algorithms such as GBDT. This paper not only improves the evaluation system, proposes a new method, but also compares different ways to deal with imbalanced dataset when they are combined with our model.

## METHODS

We propose a method, an AutoEncoder enhanced LightGBM (AEELG). The method involves data preprocessing to standardize data, AutoEncoder model to reconstruct the data feature and LightGBM to detect anomaly data from imbalanced dataset. The main process of the method is illustrated in Fig. 1.

In anomaly detection, the processing and evaluation of imbalanced dataset is one of the most challenging problems. For example, in a dataset with 100 records, there are 95 positive samples and only five negative samples. Even if all negative samples are predicted

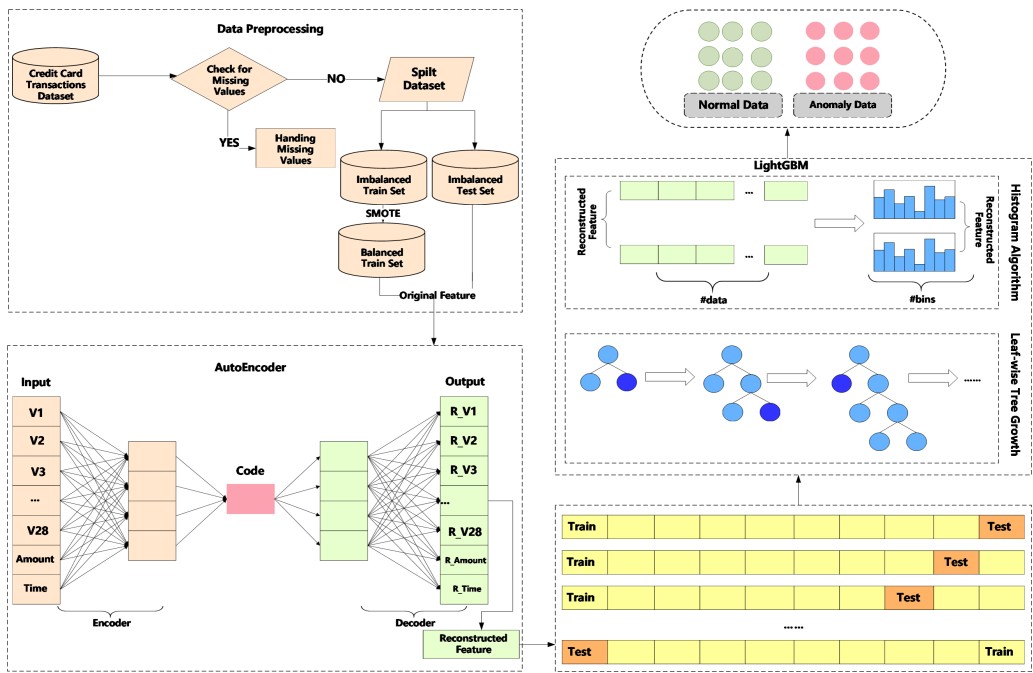

**Figure 1** Diagram of credit card detection process using AEELG.

as positive samples, the accuracy of the model would still be 95%. However, in actual application, we hope that the method can filter out the samples with a small number. Therefore, in view of the imbalance problem of credit card historical dataset, this paper adopts SMOTE algorithm to balance the training set, which is an improved scheme based on the random oversampling algorithm. To ensure the accuracy and usability of the model, the testing set is still an imbalanced dataset.

AutoEncoder is a deep neural network in the field of unsupervised learning, which can automatically learn features from unlabeled data and reconstruct input information. It can give a better feature description than the original data, and has a strong feature-learning ability. When the AutoEncoder algorithm is used for anomaly detection, it needs to put the normal data into the training set. The AutoEncoder learns the encoding format of "normal data" and uses this format to encode and decode new datasets provided to it. If the error between the decoded dataset and the input dataset falls within a certain range, it indicates that the input dataset is "normal", otherwise the input dataset is "abnormal". Aiming at the high-dimensional features of credit card and other financial data, the AutoEncoder algorithm can solve the problems of data compression and important feature extraction. AutoEncoder model, as an unsupervised learning method, has the following shortcomings when dealing with imbalanced datasets:

1. AutoEncoder model is not designed for imbalanced datasets. Its main goal is to achieve data dimensionality reduction and feature extraction. Therefore, when AutoEncoder is adopted to process imbalanced datasets, it is necessary to adjust AutoEncoder according

to specific problems, such as setting appropriate loss functions, network layers, node numbers, *etc.*

2. AutoEncoder model cannot directly perform classification tasks and needs to be combined with other classifiers, such as SVMs, decision trees, or neural networks, to use feature output from AutoEncoder. This will increase additional computational overhead and may lead to weight adjustment issues among multiple models.

3. For some complex imbalanced datasets, the performance of AutoEncoder model may be limited by uneven data distribution. Due to the small number of minority class samples, AutoEncoder model may focus more on feature extraction from majority class samples during the learning process, leading to a decrease in the ability to identify minority class samples.

4. The training process of the AutoEncoder model is unsupervised and does not consider differences between different classes in the dataset, which may lead to poor performance on classification tasks.

5. AutoEncoder model is sensitive to noise and outliers. They may be treated as normal data during training. Thereby data cleaning and processing are required to further improve the robustness of the AutoEncoder model.

Due to above shortcomings, we adopts SMOTE algorithm to balance the training set to achieve better results. SMOTE is a popular approach to balance imbalanced datasets. It works by generating synthetic samples of minority classes based on the k-nearest neighbors algorithm. The resulting dataset will have a more balanced distribution of class labels, which can improve the performance of machine learning models trained on the dataset. The basic steps of SMOTE are as follows:

1. Identify the minority class samples in the dataset.
2. For each minority class sample, randomly select k nearest neighbors from the same class.
3. Generate new synthetic samples by interpolating between the minority class sample and its k nearest neighbors.
4. Add the new synthetic samples to the dataset.

LightGBM (*Ke et al., 2017*) is a calculation framework for gradient boosting trees, which supports efficient calculations in high-dimensional features and massive data scenarios. There are many computing frameworks for gradient boosting trees, such as XGBoost (*Chen & Guestrin, 2016*), PGBRT (*Tyree et al., 2011*), *etc.* However, these two methods have problems in terms of computational efficiency and scalability in scenarios with high feature dimensions and large amounts of data. For example, XGBoost first sorts each feature and data, and then calculates the information gain of each feature and data point to determine the best split feature and split point. This process will takes up a lot of memory, and the process of sorting calculation also consumes a lot of time. Therefore, this paper uses the calculation framework of LightGBM. Specifically, LightGBM uses histogram algorithm to bucket the original data for continuous data, and methods such as the growth of leaf nodes with depth restrictions to achieve the goals of accelerating calculations and reducing memory consumption.

---

**Algorithm 1** The algorithm of anomaly detection based on AEELG

---

**Input**  Credit card fraud dataset $D$.

**Output**   test set prediction results $Y\_pred\_tst$

Standardize the dataset $D$.

Divide the standardized dataset $D$ into a training set $D\_trn$ and a test set $D\_tst$.

Use the SMOTE algorithm to oversample the training set $D\_trn$, and obtain a balanced training set $D\_bal$.

Train an AutoEncoder model on the balanced training set $D\_bal$, and obtain the encoder $f$ and decoder $g$.

Use the trained AutoEncoder model to reconstruct the features of the training set $D\_trn$ and the test set $D\_tst$:

  A. For each sample $x \in D\_trn$ and $D\_tst$:

   i. Use the encoder $f$ to compute the encoding result $z = f(x)$.

   ii. Use the decoder $g$ to compute the reconstruction result $x' = g(z)$.

  B. Obtain the reconstructed training set $D\_rec\_trn$ and test set $D\_rec\_tst$.

Train a LightGBM model on the reconstructed training set $D\_rec\_trn$ and original target variable $Y\_trn$, and obtain the training set prediction results $Y\_pred\_trn$.

Use the trained LightGBM model to predict on the reconstructed test set $D\_rec\_tst$, and obtain the test set prediction results $Y\_pred\_tst$.

**return** $Y\_pred\_tst$

---

The proposed method of AEELG can effectively address the high-dimensional and imbalanced characteristics of financial data. On the one hand, AutoEncoder can effectively extract the features of the dataset, especially the important features of the positive samples. On the other hand, as an integrated algorithm, LightGBM also has a good effect on the training and prediction of large-scale datasets. Besides, it can implement the traditional GBDT model, as well as the Gradient-based One-Side Sampling (GOSS) model and exclusive feature bundling (EFB) model. Therefore, the AEELG includes AutoEncoder +GBDT (AEELG-G), AutoEncoder +GOSS (AEELG-S), AutoEncoder +EFB (AEELG-E).

## EXPERIMENT

### Dataset and preprocessing

Two public datasets obtained from Kaggle are used to verify our method. The first dataset is the ULB Machine Learning Group's credit card fraud detection dataset, which contains transaction data from European credit card holders during two days in September 2013. Out of 284,807 transaction records, there are a total of 492 fraud records. This dataset is commonly used for research on fraud detection algorithms due to its highly imbalanced nature. The second dataset is Banco Santander's transaction prediction dataset. It is composed of 200,000 data samples, including 20,098 positive samples. This dataset is also imbalanced and is often used for research on imbalanced classification algorithms.

For the sake of privacy protection, the first dataset has been processed by principal component analysis (PCA) dimensionality reduction, which removes the background

**Table 1 The feature list of the credit card fraud detection dataset.**

| No | Field name | Data type | Field description |
|---|---|---|---|
| 1 | Time | Float | The number of seconds that have elapsed between this transaction and the first transaction in the dataset |
| 2–29 | V1-V28 | Float | Principal component data |
| 30 | Amount | Float | Transaction amount |
| 31 | Class | Int | In the case of fraud, the value is 1, otherwise it is 0 |

**Table 2 The feature list of the Santander's transaction prediction dataset.**

| No | Field name | Data type | Field description |
|---|---|---|---|
| 1 | ID_Code | Int | Unique identifier of the customer |
| 2 | target | Int | In the case of fraud, the value is 1, otherwise it is 0 |
| 3-202 | var_0-var_199 | Float | Principal component data |

information of the characteristic information of the original data. The data in the first dataset consists of 31 numerical features. These 31 features are divided into four categories: "V1-V28" is the principal component feature obtained by PCA, the "Time" feature represents the number of seconds between the current transaction and the first transaction in the dataset, and the "Amount" feature represents each sample transaction amount, the "Class" feature is the category label of the transaction, which is represented by 1 and 0. 1 means that the sample is fraudulent, and 0 means the sample is normal. The specific attribute descriptions are shown in Table 1. The second dataset includes three categories data. 'var_0-var_199' are 200 numerical features, 'ID_code' is a character type ID column, and the 'target' feature is a binary target label represented by 1 and 0. 1 means that the sample is fraudulent, and 0 means the sample is normal. The specific attribute descriptions are shown in Table 2.

## Evaluation metrics

Aiming at the characteristics of the imbalance of the dataset, our experiments not only use three traditional experimental metrics: precision (P), recall (R) and accuracy (ACC), but also add other four comprehensive evaluation metrics: F-measure(F), AUC, Matthew's correlation coefficient (MCC) (*Chicco & Jurman, 2020*) and balanced classification rate (BCR).

The calculations of the above metrics all involve the confusion matrix, so the confusion matrix is introduced first. Assuming that the sample set only contains positive and negative samples, the abnormal sample in this paper is the positive type, and the normal sample is the negative type, then the confusion matrix can be expressed as shown in Table 3.

The definition of the specific confusion matrix TP, FP, FN, TN is as follows:

- True positive (TP): The total number of samples that predict abnormal data as abnormal data;
- True negative (TN): The total number of samples that predict normal data as normal data;

**Table 3  Confusion matrix.**

| | Actual:positive (label=1) | Actual:negative (label=0) |
|---|---|---|
| Predicted:positive (label=1) | True positive(TP) | False positive(FP) |
| Predicted:negative (label=0) | False negative(FN) | True negative(TN) |

- False positive (FP): The total number of samples that predict normal data as abnormal data;
- False negative (FN): The total number of samples that predict abnormal data as normal data.

1. Precision refers to the ratio of the total number of samples correctly identified as abnormal to the total number of samples that are correctly identified as abnormal, that is the correct detection ability of the response model for abnormal data, which is calculated as Eq. (1):

$$P = \frac{TP}{TP+FP}. \tag{1}$$

2. R refers to the ratio of the total number of correctly identified abnormal samples to the number of true abnormal samples, that is the ability of the response model to identify abnormal data, calculated by Eq. (2):

$$R = \frac{TP}{TP+FN}. \tag{2}$$

3. ACC represents the proportion of correctly predicted samples in the total number of samples. Its calculation formula is as follows Eq. (3):

$$ACC = \frac{TP+TN}{TP+TN+FP+FN}. \tag{3}$$

4. F is the weighted harmonic average of precision and recall, that is, the model's comprehensive classification effect on positive and negative samples. Only when the precision and recall are both excellent, the value of F-measure is high. However, if the positive and negative samples are imbalanced, it will be affected by the larger sample size. It can be calculated according to Eq. (4):

$$F = \frac{2 \times P \times R}{P+R}. \tag{4}$$

5. AUC is defined as the area under the Receiver Operating Characteristic (ROC) curve, which comprehensively represents the classification effect of the model. The value range of AUC is [0,1]. The closer the AUC value is to 1, it indicates that the classification effect of the model is better.

6. MCC is a comprehensive metrics that measures the classification results of imbalanced data. Although both F-measure and AUC are comprehensive metrics, imbalanced datasets are usually affected by categories with large sample sizes, so we use the comprehensive metrics Matthews correlation coefficient for imbalanced datasets here. MCC is calculated by Eq. (5).

$$MCC = \frac{TP \times TN - FP \times FN}{\sqrt{(TP+FP)(TP+FN)(TN+FP)(TN+FN)}}. \tag{5}$$

7. BCR is the mean value of the respective prediction accuracy rates in the positive sample and the negative sample. Like MCC, the label selection of positive and negative cases has no effect on the calculation of BCR, and to a certain extent, it can overcome the
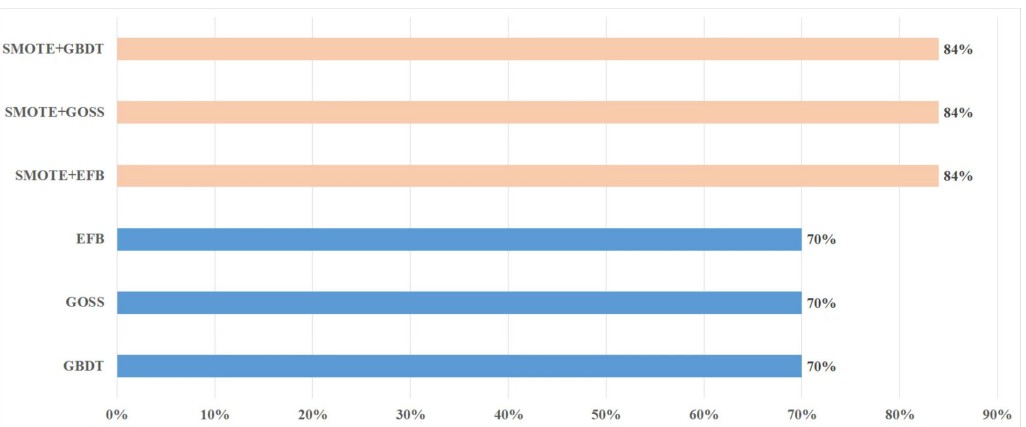

**Figure 2** Comparison of precision before and after using SMOTE.

false high of the evaluation metrics caused by the imbalance of positive and negative samples. The function is shown in Eq. (6).

$$BCR = \frac{1}{2}\left(\frac{TP}{TP+FN} + \frac{TN}{TN+FP}\right). \tag{6}$$

# RESULTS

## Performance on credit card fraud detection dataset

Aiming at the imbalance of dataset, SMOTE method is adopted to preprocess the dataset. Figure 2 is the comparison of experimental accuracy before and after using the SMOTE method. It can be seen from the figure that the precision of the experiment increased by about 14% after adopting the SMOTE method.

In the experiment, different thresholds training AutoEncoder model are selected for comparison. Table 4 shows that as the threshold increases, the precision decreases, while the recall increases, indicating an improved ability to recognize abnormal data. Due to the imbalance of the test set samples, we finally selected AutoEncoder model with a threshold of 2 through the comprehensive metric of MCC.

It can be seen from Fig. 3 that the metrics of the original GBDT, GOSS, and EFB models are not much different. Among them, EFB model outperforms the other three models in all metrics. Therefore, it can be concluded that EFB model has better comprehensive performance. Regarding specific metrics, EFB model achieves a relatively high F-measure of 70.27%, which is about 5.68% higher than that of the GBDT model, indicating that it strikes a good balance between precision and recall. The AUC value of EFB model is also relatively high, at 84.37%, which is approximately 0.84% higher than that of the GBDT model, demonstrating its superior performance in sample classification problems. Additionally, the precision and MCC of EFB model are the highest among all models,

**Table 4  Performance of the AutoEncoder model under different thresholds.**

| AutoEncoder Threshold | P | R | F | ACC | MCC |
|---|---|---|---|---|---|
| 1 | 83.94% | 15.58% | 26.29% | 97.30% | 35.49% |
| 1.5 | 71.34% | 23.11% | 34.91% | 99.01% | 48.20% |
| 2 | 56.50% | 30.55% | 39.65% | 99.43% | 53.16% |
| 2.5 | 41.06% | 38.21% | 39.58% | 99.49% | 48.64% |
| 3 | 32.52% | 42.99% | 37.03% | 99.50% | 45.04% |

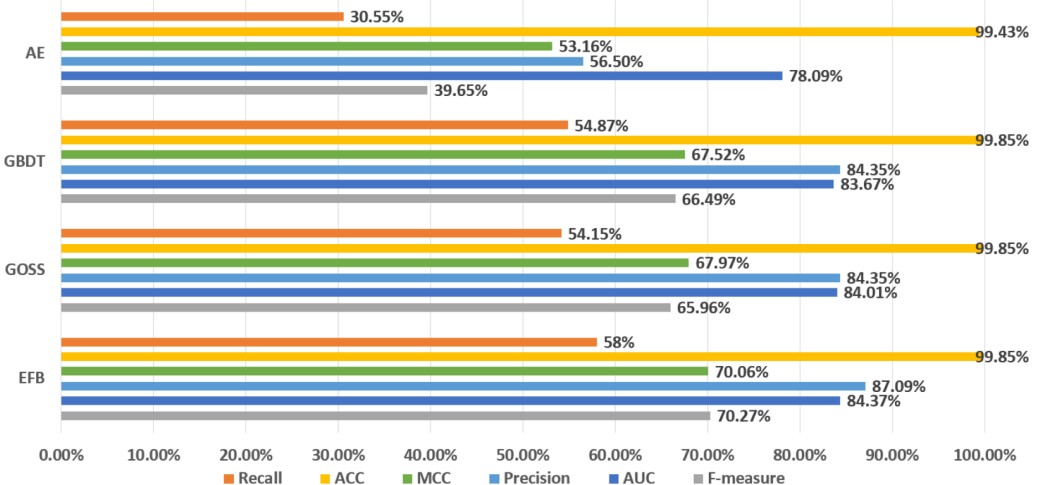

**Figure 3  Comparison of LightGBM and AutoEncoder.**

suggesting that it is more accurate in predicting positive cases and effectively avoids false positives.

In addition, AutoEncoder model performs lower in all metrics than GBDT, GOSS, and EFB models, which means that AutoEncoder model has a weaker ability to identify abnormal data and accurately detect it, while the recall of normal data is higher. In other words, since AutoEncoder model effectively extracts the features of normal data, it has a strong ability to recognize normal data. Considering that the original GBDT, GOSS, and EFB models have a strong ability to recognize abnormal data, while AutoEncoder has a strong ability to recognize normal data, we use AutoEncoder enhancing GBDT, GOSS and EFB, trying to attain the best performance. The results before and after being enhanced are shown in Fig. 4.

Comparing AEELG with the single model before enhancement, the recall of AEELG greatly improved, the precision slightly decreased, and the three comprehensive metrics of F-measure, AUC, and MCC greatly improved. Compared with the original GOSS model, the recall of the AEELG-S has increased from 54.15% to 91.74%, indicating that the hybrid model has greatly improved the ability to recognize abnormal data. However, the precision has relatively dropped from 84.35% to 68.03%, implying a decrease in the ability of the

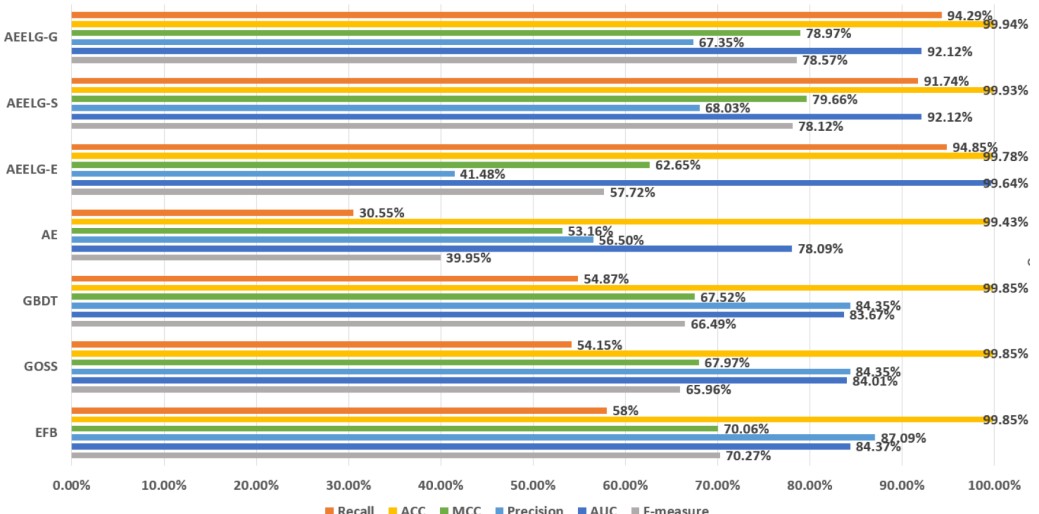

**Figure 4** Comparison of experimental results before and after being enhanced.

AEELG-S to correctly detect abnormal data. Nonetheless, other comprehensive metrics have improved. Compared with the single model, the F-measure of the AEELG-S has increased from 65.96% to 78.12%, the AUC has increased from 84.01% to 92.12%, and the MCC has increased from 67.97% to 79.66%, respectively.

In general, the method has improved all comprehensive metrics compared with the single models. The recall is up to 94.85%, which shows that the feature attributes of normal data are strengthened after reconstructing the features using AutoEncoder model. Although the precision has dropped slightly, the recall has greatly improved, which suggests the ability to identify abnormal data has greatly improved, and the three comprehensive metrics of F-measure, AUC, and MCC have also improved.

In order to discuss the predictive performance of AEELG when it is combined with different sampling methods, we used nine sampling methods to balance the dataset respectively. Among them, oversampling methods include SMOTE and ADASYN, undersampling methods include RandomUnderSampler, NearMiss, CondensedNearestNeighbour, OneSidedSelection, editednearestneighbors, and InstanceHardnessThreshold, and the mixed sampling method is Borderline-SMOTE method. According to Fig. 5, SMOTE ranked highly in all performance metrics, with a recall of 93.72% ranking third overall. Although the NearMiss algorithm ranked first in terms of recall with a high score of 89.94%, its precision and F-measure scores were both below 1%. This suggests that the algorithm misclassifies too many positive samples as negative ones when negative examples are filtered out, resulting in significant errors and misjudgments during prediction. Therefore, the performance of the NearMiss algorithm is inadequate for practical applications. Although the ADASYN algorithm improved recall by 0.52% compared to SMOTE, its F-measure score decreased by 2.35%, this results indicate that while the ADASYN algorithm successfully increased the number of positive samples, it also misclassified some negative samples as positive, resulting in a decrease in precision and F1 score. Specifically, although the

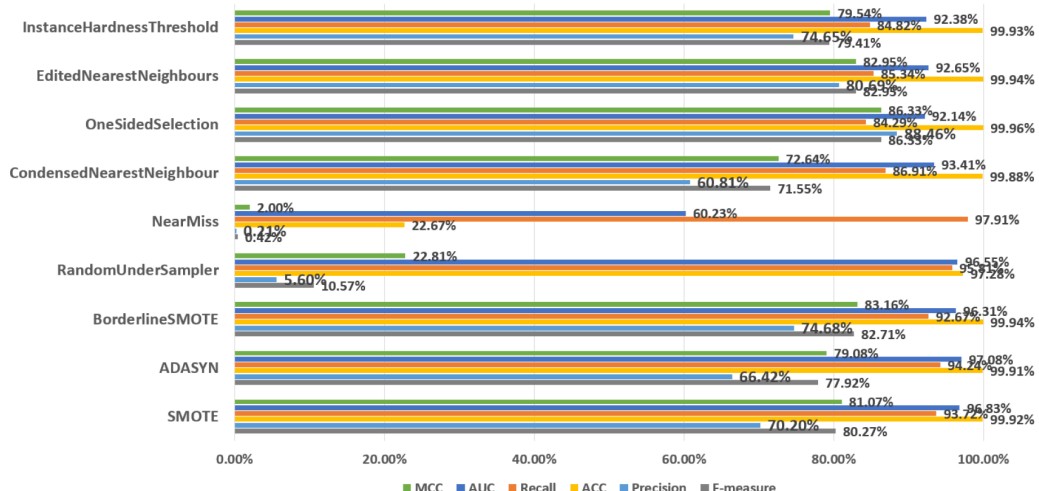

**Figure 5** Comparsion of different sampling methods.

ADASYN algorithm can improve the model's recall, it introduces additional errors, leading to an overall decrease in model performance. Furthermore, the Borderline-SMOTE algorithm, an improvement of SMOTE, can better maintain the class distribution of data and improve the classifier's performance in identifying minority classes and overall performance compared to the SMOTE algorithm, despite having a slightly lower recall rate by 1.05%. Specifically, the MCC increased by 2.09%, precision increased by 4.48%, and F-measure increased by 2.54%. These improvements suggest that the combination of Borderline-SMOTE and AEELG leads to promising performance in predicting imbalanced datasets. Overall, based on multiple performance metrics, the predictive performance of the AEELG model is maximized when it is combined with either SMOTE or Borderline-SMOTE algorithm.

In order to evaluate our method furtherly, we compared AEELG with the experimental results of reference (*Randhawa et al., 2018*) in Fig. 6, reference (*Bagga et al., 2020*) in Figs. 7, 8, 9, and 10, and reference (*Alharbi et al., 2022*) in Fig. 11 on the same dataset respectively. A hybrid model that employed a combination of AdaBoost and majority voting methods is introduced for classification in *Randhawa et al. (2018)*. Specifically, this model involves training multiple weak classifiers using the AdaBoost algorithm and assigning weights to them based on their accuracy. Subsequently, the output of all weak classifiers is processed using a majority voting scheme, and the final classification decision is determined by selecting the class with the highest number of votes. This hybrid method offers several advantages, including the ability to leverage the powerful features of AdaBoost while mitigating some of its drawbacks such as overfitting. The second article proposed two models for anomaly detection. The first model used a Pipeline to assemble multiple processing steps. Firstly, the "selectKBest" method is used to choose k optimal features from the dataset using f-regression as the scoring function. Each feature is subject to linear regression tests to calculate its impact score. Then, the preprocessed data is passed

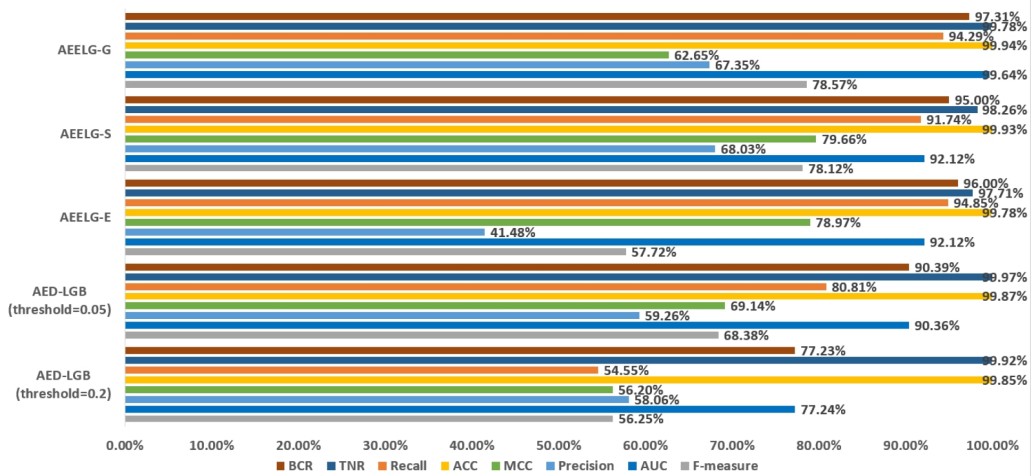

**Figure 6  Comparison between AEELG and AED-LGB.**

to a random forest classifier for classification and prediction. Both feature selection and random forest classifier techniques are employed in this model to improve the classification accuracy and avoid overfitting. The second model used ensemble learning to combine multiple base classifiers and enhance the performance of a single classifier. Specifically, this paper employed a Bagging classifier that fits base classifiers to random subsets of the original dataset and aggregates their predictions through voting or averaging to generate a final prediction. By introducing randomness, the Bagging classifier reduced the variance of black-box classifiers, such as decision trees, eventually created an improved ensemble model. The third paper proposed an anomaly detection method based on text2IMG conversion technology and CNN. The author uses text2IMG to convert text dataset into image and uses inverse frequency method to solve class imbalance in the conversion process. Then images with class weights are put into CNN architecture. Finally, the features extracted by CNN are applied to other machine learning classifiers for classification.

Compared with *Randhawa et al. (2018)*, the recall and the ACC of AEELG-G, AEELG-S and AEELG-E in this paper are all the highest, which means that our proposed method performs the best among majority voting methods in terms of the ability to detect anomalies. The comprehensive metric MCC of AEELG-S ranked the second with 79.66%, which is better than RF+GBT, NB+GBT, DT+NB, DT+GBT, and DT+DS, but is inferior to NN+NB. As shown in Fig. 7, for ACC, the distinction among AEELG and methods in reference is small, and all of them achieved over 99%. The results of the above comparison show that the characteristics of AutoEncoder reconstruction features can enhance the LightGBM, resulting in further improving its performance in anomaly detection.

Compared with *Bagga et al. (2020)*, the recall of our method is still the highest with 94.85%, which is higher than the 91% recall, the highest value in the reference. As shown in Fig. 9, the comprehensive metric MCC of our method ranks third with 79.66%, which is inferior to Pipelining and Ensemble Learning. For another comprehensive metric BCR shown in Fig. 10, our method performs best. The BCR values of AEELG-S is 97%, which

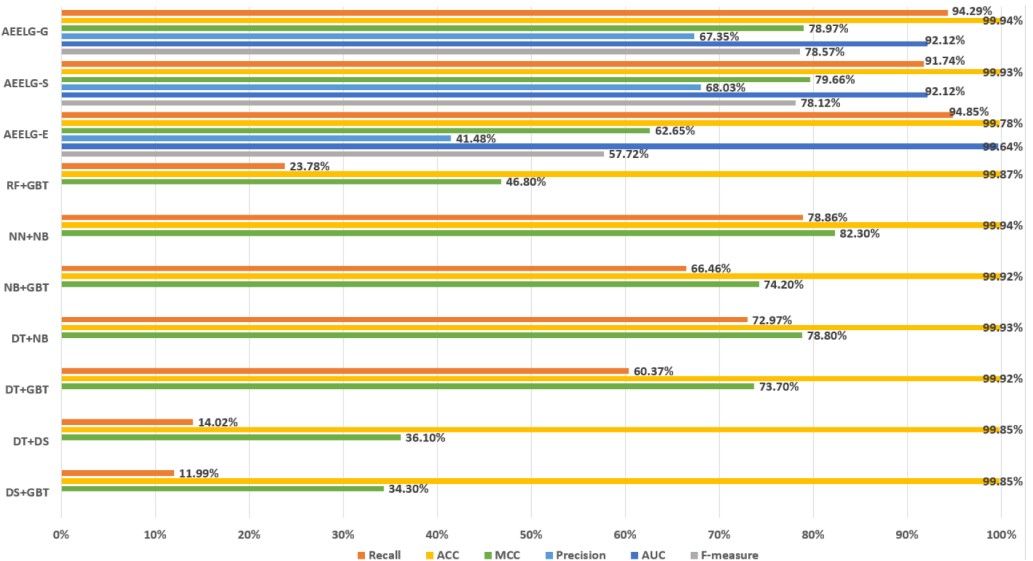

**Figure 7** Comparison between AEELG and majority voting combination methods.

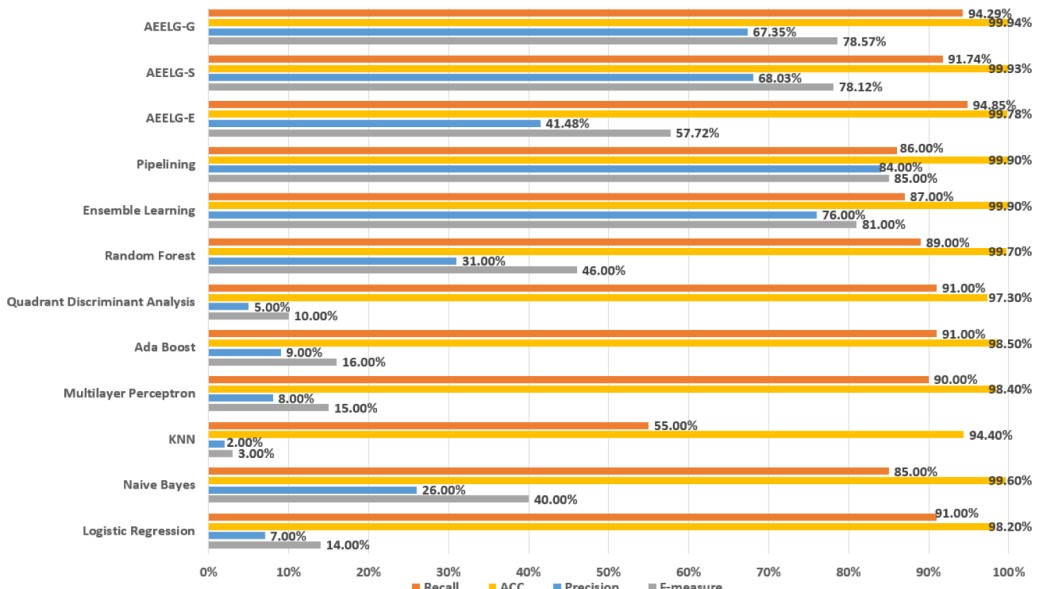

**Figure 8** Comparison between AEELG, Pipelining, and Bagging Methods.

is the highest among all algorithms. In addition, AutoEncoder adds constraints and restricts on learning useful features, especially the features of positive samples, thereby strengthening the characteristic attributes of normal data are strengthened. Therefore, the final comprehensive metrics of the AEELG have been improved.

As shown in Fig. 11, compared with *Alharbi et al. (2022)*, AEELG-G, AEELG-S and AEELG-E have the highest recall and ACC. AEELG-G and AEELG-S also surpassed machine

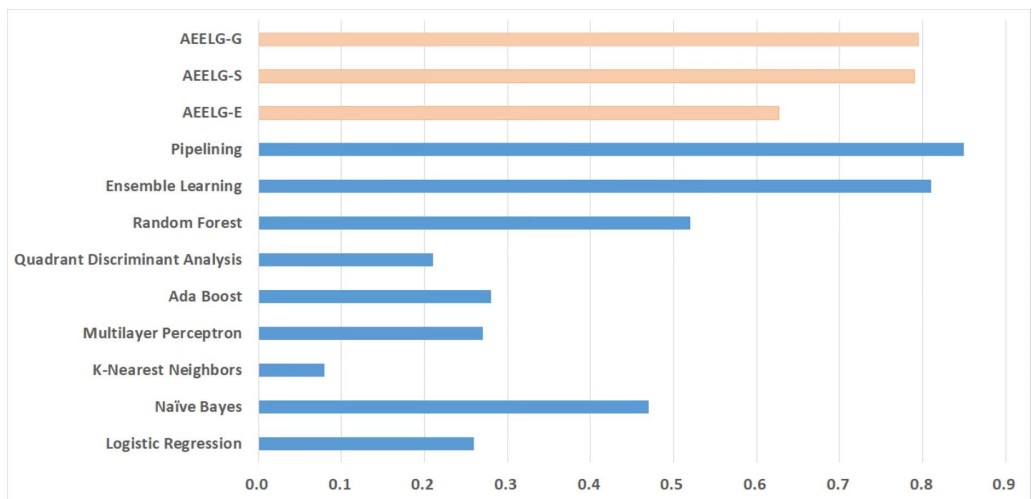

**Figure 9** Comparison between AEELG, Pipelining, and Bagging Methods in MCC.

learning classifiers using CNN to extract features in the other three metrics. Although the precision of AEELG-E is lower than that of CNN+KNN and CNN+Bagged-Boost Ensemble method. AEELG-E's F-measure is only 0.08% lower than CNN+rough-KNN. Meanwhile it owns the highest recall among all methods. Our method uses AutoEncoder for enhanced features, which has a better ability to detect real anomalies in credit card fraud detection than using CNN for reconstructed features. It can be seen from the classification results of CNN and AutoEncoder that F-measure and precision of AutoEncoder are higher than those of CNN but recall is lower, which indicates that AutoEncoder can capture features in majority class and has good representation ability for majority class but it is less sensitive to a minority class than CNN. When SMOTE is adopted to balance the dataset, the detection ability of AEELG model is much stronger than the classification methods using CNN for features extraction. Once again it proves the superiority of AEELG algorithm and the effectiveness of SMOTE algorithm when it is combined with our model.

## Performance on Santander's transaction prediction dataset

The Santander transaction prediction dataset contains 200 features with low correlation, making it difficult to achieve good classification performance using a single classifier. As shown in Fig. 12, the top two F-measure scores are achieved by AEELG-G and AEELG-S at 53.68% and 53.32%, respectively. It indicates that these two models have good overall classification capabilities. The top two recall scores are also achieved by AEELG-G and AEELG-S at 54.74% and 69.51%, which suggests that they have better sensitivity and can detect more anomalous samples. The top two AUC scores are achieved by AEELG-G and AEELG-S at 89.85% and 89.05%, and the top two MCC scores are achieved by them at 50.70% and 48.33%. This indicates that both AEELG-G and AEELG-S have excellent ability to distinguish between positive and negative samples and can perform well on imbalanced datasets. In particular, the AEELG-G model outperforms AEELG-S in terms of MCC score, indicating that it can more accurately predict positive and negative samples while reducing

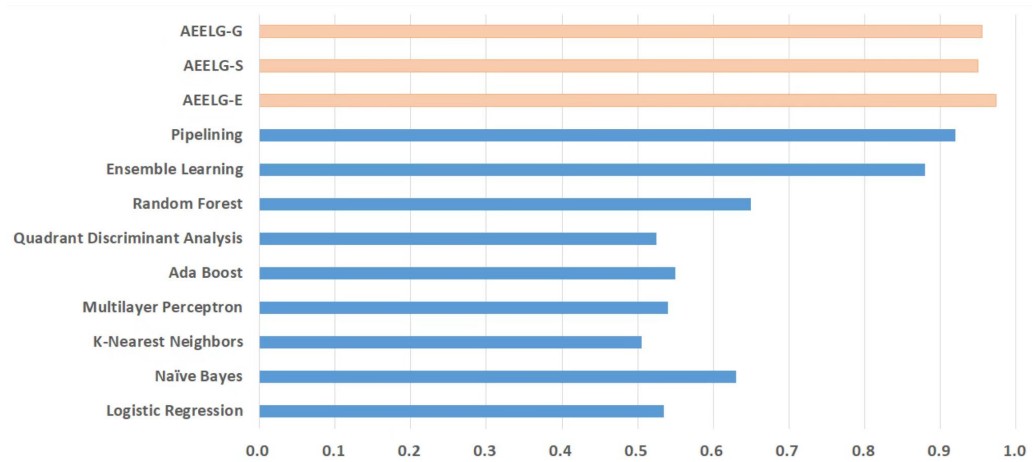

**Figure 10  Comparison between AEELG, Pipelining, and Bagging Methods in BCR.**

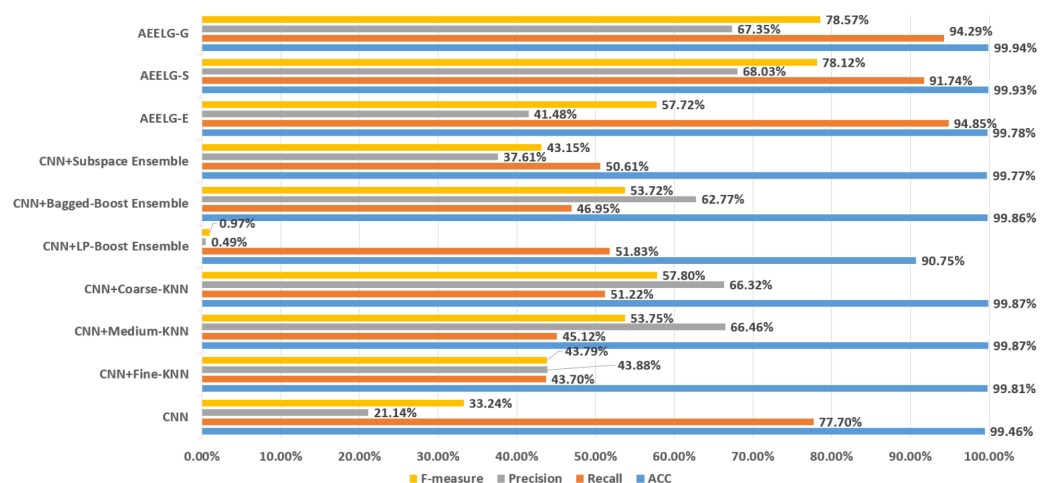

**Figure 11  Comparison among AEELG and CNN enhanced machine learning methods.**

misdiagnosis and missed diagnosis rates, thereby demonstrating superior classification performance. Overall, the AEELG algorithm exhibits good classification performance and has better ability to distinguish between positive and negative samples.

## CONCLUSION

This article proposes an AutoEncoder enhanced LightGBM approach for anomaly detection in high-dimensional and large-scale financial data. In order to deal with imbalanced data, we discuss the performance of SMOTE, Borderline-SMOTE and other seven sampling methods when they are combined with our proposed method. These nine sampling methods are evaluated on multiple comprehensive metrics. The experimental results indicate that the SMOTE sampling method and Borderline-SMOTE, one improvement of

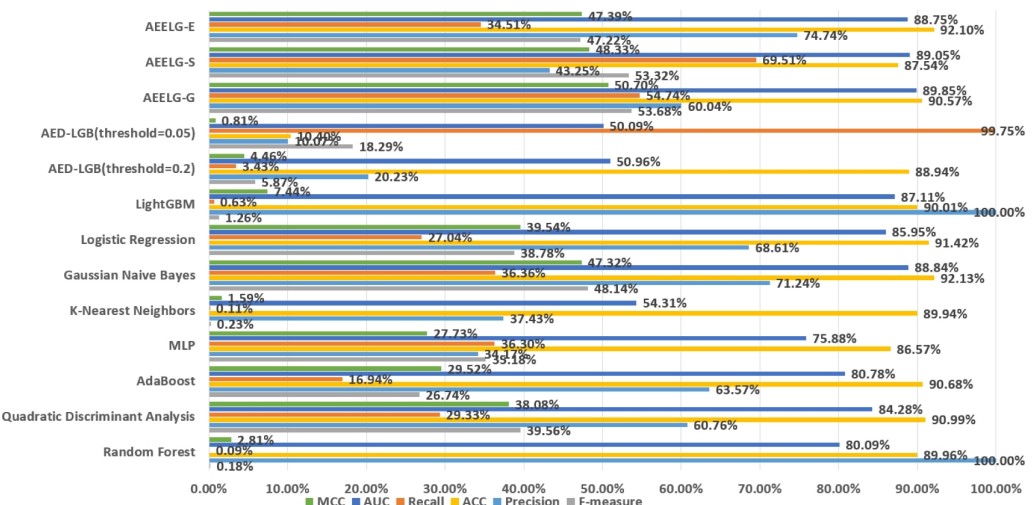

**Figure 12** Comparison AEELG with other machine learning methods.

SMOTE, can effectively solve the problem of data imbalance. Compared to the 22 methods mentioned in the reference literature, our approach achieved significant improvements in most evaluation metrics. Additionally, based on the experimental results conducted on the Santander Bank transaction records dataset, it can be concluded that the proposed method, AEELG, outperforms other classifiers in detecting anomalies and exhibits better performance in handling high-dimensional data. These research findings are of great significance for financial institutions to develop more accurate and reliable fraud detection systems. In the future, we will be working to improve our model on the metric of precision. Although the comprehensive metrics of our method have performed well, the precision still decrease compared with some single models. We are planning to explore a newer model combining another unsupervised learning method—generative adversarial nets (GANs), whose ultimate goal is to get a high-quality automatic generator and a classifier with strong judgment ability.

### Funding

This work was supported by the National Natural Science Foundation of China (72101033, 71831001), the Beijing Natural Science Foundation Project (KZ202210037046), the Beijing Key Laboratory of Intelligent Logistics Systems (BZ0211), the Construction Project of Innovation Group of Southern Marine Science and Engineering Guangdong Laboratory (Zhuhai) (311020012), the Research Program of Beijing Municipal Education Commission (KM201910037001, KM202010037002), and the Excellent Science and Technology Innovation Team Project in Tongzhou District (CXTD2023010). The funders had no role in study design, data collection and analysis, decision to publish, or preparation of the manuscript.

## Grant Disclosures

The following grant information was disclosed by the authors:

National Natural Science Foundation of China: 72101033, 71831001.

Beijing Natural Science Foundation Project: KZ202210037046.

Beijing Key Laboratory of Intelligent Logistics Systems: BZ0211.

The Construction Project of Innovation Group of Southern Marine Science and Engineering Guangdong Laboratory (Zhuhai): 311020012.

Research Program of Beijing Municipal Education Commission: KM201910037001, KM202010037002.

Excellent Science and Technology Innovation Team Project in Tongzhou District: CXTD2023010.

## Competing Interests

The authors declare there are no competing interests.

## Author Contributions

- Lianhong Ding conceived and designed the experiments, prepared figures and/or tables, and approved the final draft.
- Luqi Liu performed the experiments, performed the computation work, prepared figures and/or tables, and approved the final draft.
- Yangchuan Wang performed the experiments, authored or reviewed drafts of the article, and approved the final draft.
- Peng Shi conceived and designed the experiments, prepared figures and/or tables, and approved the final draft.
- Jianye Yu analyzed the data, authored or reviewed drafts of the article, and approved the final draft.

## Data Availability

The first credit card fraud detection data is available at Kaggle and Zenodo:

- https://www.kaggle.com/datasets/mlg-ulb/creditcardfraud.

- Luqi Liu. (2022). Credit Card Fraud Detection [Data set]. Zenodo. https://doi.org/10.5281/zenodo.7395559.

The second Santander customer transaction data is available at Kaggle and Zenodo:

- https://www.kaggle.com/competitions/santander-customer-transaction-prediction.

- Liu. (2023). Santander Customer Transaction Prediction [Data set]. Zenodo. https://doi.org/10.5281/zenodo.10212042.

The code is available at GitHub and Zenodo:

- https://github.com/ardy657/An-AutoEncoder-enhanced-LightGBM-.git

- ardy657. (2022). ardy657/An-AutoEncoder-enhanced-LightGBM-: v1.0.0 (v1.0.0). Zenodo. https://doi.org/10.5281/zenodo.7387762.

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
