# Peer review of "An AutoEncoder enhanced light gradient boosting machine method for credit card fraud detection"

_PeerJ Computer Science, doi:10.7717/peerj-cs.2323_

## Round 0.1 · original submission · Major Revisions

- The contributions of the paper need to be highlighted in the introduction.
- The research gap needs to be addressed well in the introduction.
- Sampling methods should be discussed to solve the imbalanced data.

Reviewer 1 ·

Basic reporting

Paper review by Srikanth Tammina – An Auto encoder enhanced LightGBM method for credit card fraud detection

Abstract – Abstract is clear and good. The proposed methodology is already worked by many researchers and the case work proposed here is weak. Proofreading is needed.

Introduction - Overall good framework in describing what the research work is about. Re-write phases as there are grammatical errors. Citations are not properly cited. In related work, authors can add more historical literature in the research and focus on recent research happened in this field i.e., 2021/2022. Also, authors can work on to expand figure 1 more clearly.

Experimental design

Methods/Model –The experimental design is weakly designed architecture but rightly designed to solve the problem described in the abstract. The framework designed is relevant and all formulas are neatly denoted. Dataset is rightly chosen for this problem and relevant. Grammatical check in experiments section. I suggest authors to add complexity in the notation of Big (O) to understand how best your work has reduced the computational complexity. Authors can use more complicated algorithms to solve this problem and use vast dataset to understand the efficacy of the SMOTE+GBDT. Also, SMOTE has issues in current ML problems. Advise to use alternative to SMOTE.

Validity of the findings

Since the dataset is less I believe authors have high accuracy. Increase the dataset and use more granular data to understand how robust your models are.

Additional comments

Conclusion – Can be re-phase and written more clearly. Can add more details to conclusion. Authors should also focus on grammar of the literature in the whole of their research paper. Also, the algorithm and proposed methodology is commonly used in the industry. Nothing original and substantial work done.

Annotated reviews are not available for download in order to protect the identity of reviewers who chose to remain anonymous.

Reviewer 2 ·

Basic reporting

figure 1 is in need to be more clearer-utilize the space to the left to enlarge the figure
line 122,124, "Taha A A and Malebary S J proposed...", "Tang et al.(Tang et al., 2020) " is this a correct way to cite authors
give some space between words (line 129, 164 as an example)

though you said that "Another major challenge in credit card transaction fraud detection is the imbalance of datasets.", you only provided few references/cases/examples of prior studies that talked about this issue. you need to elaborate this section to reveal how others handle this issue

you need to fix this "Algorithm 1 The algorithm of anomaly detection based on AEELG is shown in Algorithm 1"

you compared you results with others' work such as Bagga, but you did not mention their work in LR. you need to know what they have done

Experimental design

it is fairly clear. but they did not explain how SMOTE balanced the dataset- at least tell us its technique

Validity of the findings

done

Reviewer 3 ·

Basic reporting

1- Literature references are insufficient there is a lack of recent citations.
The introduction and related work section are not well written the issues and gap
2- Don’t use abbreviations in the title, write the full name of the method
3- In the abstract, the results should be expressed in terms of the improvement ratio, the obtained results have no meaning without comparison.
4- The introduction section is too general and not well written. The conclusion of the prior research related to the topic should be presented. The motivation issue and the gap in the existing research should be highlighted.
5- There is a lack of recent references in the area. You should survey the recent related work. Most of the references are too old.
6- Figure citations in the text should be revised
7- Algorithm 1 should be rewritten using pseudocode

Experimental design

1. The problem that the research tries to solve is not clear. The gap is not highlighted.
2. The limitations of prior studies and how they fail to address the gap should be clearly stated.
3. There is a lack of deep investigation of the related work.
4. The author claimed that the proposed method of AEELG can well solve the high-dimensional and imbalanced characteristics of financial data. However, the data set used is already compressed using PCA and is not highly dimensional anymore. In addition, it is not clear how AEELG solved the problem of the imbalance dataset. Many authors used SMOTE technique to balance the dataset. It is not clear how these two problems were solved.

Validity of the findings

In the results authors need to explain why AutoEncoder+EFB has same exact results like AutoEncoder+GBDT.
What is the pipelining in Figures 7 and 8 ?

---

## Round 0.2 · Major Revisions

The authors should justify the similarity/differences with this paper: Haichao Du et al. AutoEncoder and LightGBM for Credit Card Fraud Detection Problems (https://www.mdpi.com/2073-8994/15/4/870).

You must highlight the novelty of this paper in terms of applying different models and obtaining better results.

This paper should be cited and used in the comparison & discussion of the results.

Other minor issues such as typos must be fixed.

Reviewer 1 ·

Basic reporting

reporting has been improved. Accept.

Experimental design

After carefully evaluating the research paper, I found another similar research paper published in April 2023, which solves the same problem with the same model architecture.

Please refer to the below paper -
https://www.mdpi.com/2073-8994/15/4/870
(Haichao Du et al. AutoEncoder and LightGBM for Credit Card Fraud Detection Problems)
Attached is the link to the paper that solved the credit card classification problem using the same Autoencoder with the LGBM model that this paper used. They also used SMOTE and all the evaluation metrics similar to this paper to calculate the effectiveness of the model and improve the imbalanced datasets. I also do not see any references made to this paper too.

I advise authors to solve the credit card classification problem by setting a new experimental design after careful research of past research papers and then coming up with the original experimental design. SMOTE + Autoendoder + LightGBM has already been designed by others similar to the above. Also, the dataset is the same in this research paper and the above-referenced research paper. I don't see any novelty in this research paper. Advise authors to enhance their work by adding novelty to the experimental design and rigorous investigation in past works.

Experimental design is not original primary research and fails the standards set by the journal.

Validity of the findings

Accept.

Reviewer 3 ·

Basic reporting

Authors have addressed all the comments raised in the first-round revision. However, there are still some linguistic issues, such as typos. Authors should carefully revise the language.

Experimental design

Authors have addressed all the comments raised in the first round revison

Validity of the findings

Authors have addressed all the comments raised in the first round revison

---

## Round 0.3 · accepted · Accept

Congratulations, you paper is ready for publication.

Reviewer 1 ·

Basic reporting

Accept

Experimental design

Accept

Validity of the findings

Accept